# Research on the evolution and driving forces of the manufacturing industry during the "13th five-year plan" period in Jiangsu province of China based on natural language processing

**Shiguang Shen[1], Chaoyang Zhu[1], Chenjing Fan[1]\*, Chengcheng Wu[1], Xinran Huang[1], Lin Zhou[2]**

**1** College of Landscape Architecture, Nanjing Forestry University, Nanjing, China, **2** Institute of Industrial Economics of CASS, Beijing, China

\* fanchenj@qq.com

**Funding:** The research was funded by NO.2018YFD1100105 National Key Research and Development Program of China, NO.2020NFUSPITP0910 Innovative training program for College Students of Nanjing Forestry

## Abstract

The development of China's manufacturing industry has received global attention. However, research on the distribution pattern, changes, and driving forces of the manufacturing industry has been limited by the accessibility of data. This study proposes a method for classifying based on natural language processing. A case study was conducted employing this method, hotspot detection and driving force analysis, wherein the driving forces industrial development during the "13th Five-Year plan" period in Jiangsu province were determined. The main conclusions of the empirical case study are as follows. 1) Through the acquisition of Amap's point-of-interest (POI, a special point location that commonly used in modern automotive navigation systems.) data, an industry type classification algorithm based on the natural language processing of POI names is proposed, with Jiangsu Province serving as an example. The empirical test shows that the accuracy was 95%, and the kappa coefficient was 0.872. 2) The seven types of manufacturing industries including the pulp and paper (PP) industry, metallurgical chemical (MC) industry, pharmaceutical manufacturing (PM) industry, machinery and electronics (ME) industry, wood furniture (WF) industry, textile clothing (TC) industry, and agricultural and food product processing (AF) industry are drawn through a 1 km× 1km projection grid. The evolution map of the spatial pattern and the density field hotspots are also drawn. 3) After analyzing the driving forces of the changes in the number of manufacturing industries mentioned above, we found that manufacturing base, distance from town, population, GDP per capita, distance from the railway station were the significant driving factors of changes in the manufacturing industries mentioned above. The results of this research can help guide the development of manufacturing industries, maximize the advantages of regional factors and conditions, and provide insight into how the spatial layout of the manufacturing industry could be optimized.

University, NO.20CJL004 National Social Science
Fund of China.

**Competing interests:** The authors have declared
that no competing interests exist.

## Introduction

China's economic development, especially in its manufacturing sector, has received global
attention [1]. The implementation of the "One Belt, One Road" initiative has accelerated the
development of China's manufacturing industry [2, 3], which has acquired a reputation for
being a "world factory." The manufacturing industry is the mainstay of China's national econ-
omy; thus, China's rapid economic growth has been largely driven by the development of its
manufacturing industry. Two main patterns have been noted during the rapid expansion of
China's manufacturing industry: the manufacturing industry has continued to grow both in
terms of quality and quantity [4], and manufacturing activities are concentrated in a few
regions [5]. However, the spatial patterns of manufacturing vary in different regions because
of differences in human capital, innovation capacity, technology absorptive capacity, and capi-
tal policies [6]. The coastal provinces in eastern China have advantageous geographical loca-
tions and greater economic capital and technological development and thus have played a
critical role in China's economic development.

Many studies have examined the evolution of the spatial patterns of manufacturing and its
driving force in China [7–10]. In terms of the evolution studies, data mainly have been col-
lected from various economic censuses [11, 12], statistical yearbooks [13, 14], large and
medium-sized enterprise databases [15]. However, the data obtained by these methods often
lack accurate spatial coordinate information. This is problematic given that spatial changes in
the manufacturing industry are mainly reflected by microgeographical openings and closures
of individual companies. According to previous studies, some of the drivers of manufacturing
include Economy (wage and land price [16], GDP per capita, and R&D internal expenditure
[17], and the entry and exit of closely related industries [18]), Population (quality of labor and
number of workers [19], Environment (environmental regulation [17]), Policy (size of devel-
opment zone [19], industrial policies mentioned by provincial government [20]), and Location
(distance to the Great Lakes and the East coast [16]). However, because of the lack of high-res-
olution data, research on driving forces has been conducted at provincial and prefecture-level
city scales.

Point-of-interest (POI, a special point that is commonly used in modern automotive navi-
gation systems) with accurate geospatial information was the main approach used for naviga-
tion years ago. With the development of data collection and processing technology, there has
been increased interest in POI information mining and application [21]. XU Dong [22] used
POI data to analyze the spatial characteristics of leisure tourism in Nanjing. RAN Zhao [23]
used POI data for catering, leisure, and medical care to study spatial patterns in the consumer
service industry in Changsha. LAN Zhenjia [24] used POI data for elementary schools, traffic
stations, and residential areas to evaluate the status of elementary education resources in east-
ern cities in China. CUI Zhenzhen [25] used POI data of public service facilities to evaluate
urban life convenience. Fan Chenjing [26] used POI data, remote sensing data and data from
urban statistical yearbooks to evaluate ecological spatial planning. CHEN Shili [27] used POI,
floating car GPS, and road network data to identify different functional zones. ZHAO Yunhan
[28] used POI, Tencent user density, and building footprint data to extract buildings of urban
villages in Guangzhou. The types of POI data used by studies have mainly included commer-
cial facilities, living facilities, and cultural facilities. Such data are updated promptly and have
high classification accuracy. However, data types such as companies and enterprises have low
classification accuracy, and the classification system is often incomplete. Consequently, few
spatial studies of companies using POI data have been conducted.

In general, few studies have examined the evolutionary dynamics of the structure of the
manufacturing industry at the micro-level, and research on the driving forces of the spatial

elements of the industrial layout has yet to be conducted. Although POI big data can be used to study the spatial patterns of various urban facilities at the theoretical and technical level, current research on POI has been mostly focused on characterizing the patterns of spatial density in the original data. Further data mining is then carried out based on these data. In this study, a natural language processing algorithm based on existing POI research methods is used to semantically classify the names of manufacturing POI data and convert the original data into spatial pattern data. Using the framework described above, this paper uses Jiangsu, China as a case study and examines the evolution of the manufacturing industry and its driving forces. Generally, this paper establishes a set of research methods for studying the evolution of the manufacturing industry at the micro-scale. Valuable manufacturing-related data are provided, which could be used to optimize the layout of the manufacturing industry.

## Research framework

The number of POIs is almost equal to the number of facilities in the world. Therefore, the evolution of the industrial structure and its driving forces can be feasibly studied through classified POI data. The research process is divided into 4 steps. First, POI data are collected, and the knowledge base of POI industry types is constructed. Second, POI data are cleaned and classified. Third, the industrial space pattern is drawn. Fourth, the driving forces of industry evolution are analyzed.

## Industry type analysis based on natural language processing

Most recent research has focused on analyzing the evolution of industrial spatial patterns [23]; however, few studies have analyzed industrial spatial patterns after having segmented industries. For this reason, the first step in our analysis is to construct the industry type domain knowledge base. Specifically, the word segmentation tool is used to separate the POI data and summarize the high-frequency keywords related to different industries through statistical software. Next, the research objectives are classified, and the middle-class industries [29] are subdivided according to the National Economic Industry Classification [30] in the National Standard of the People's Republic of China GB/T 4754–2017. Next, the summarized keywords are classified according to different industry types. A knowledge base of industry type domains is formed, which is expressed as $(d_1,c_1,l_1),(d_2,c_2,l_2),(d_3,c_3,l_3)\ldots(d_i,c_i,l_i)$, where $d_i$ represents the i-th semantically segmented keyword, $c_i$ represents the i-th keyword category, and $l_i$ represents the length of the i-th keyword.

## Text classification algorithm based on Bayesian probability

To transform text into a data structure that can be processed by a computer, the text needs to be divided into semantic units. In this paper, the naive Bayes classifier with the forward maximum matching method is used. The naive Bayes classifier gives a probability evaluation to each successfully matched object in the POI name $S$. When a new POI name $S$ is passed into the classifier, it outputs the most likely category c. Formula 1 shows the algorithm:

$$\hat{c} = \arg\max_{c \in C} P(c|d) = \arg\max_{c \in C} \frac{P(d|c)P(c)}{P(d)} = \arg\max_{c \in C} \log P(c) + \sum_{i \in positions} \log P(w_i|c) \quad (1)$$

For each continuous Chinese character keyword $w_i$ in $S$, all keywords of category c in the knowledge base training data set are located, and the number of occurrences of keyword $w_i$ in $S$ (category c) is measured. Next, the total number of keywords in the knowledge base category c is calculated, as well as the ratio between the two above, and $P(w_i|c)$ is obtained. For example,

on the length L, if the probability P(w$_i$|c) of a certain category c is high, then the POI enterprise name is determined as a certain type c industry. If there are scenarios with the same classification probabilities, the L-1 length is calculated until specific industry categories are classified. Finally, the above classification results are combined with POI xy-coordinates in the geographic information database.

## Measurement of manufacturing spatial pattern evolution based on the classification results

The hotspot detection method based on the density surface proposed by Zhang Haiping is used [31]. First, the kernel density estimation method is used to construct a density surface model for manufacturing POIs. After the heat surface is generated, the values of the density surface are generated, which are referred to as the focus statistics of the raster data. The neighborhood maximum values are then statistically analyzed to construct a surface of extreme pixel values. Thereafter, map algebra is used to perform an overlay calculation between the density and the extreme surfaces to identify the local maximum values. All the local maxima of the constructed density surface are then extracted by the POIs, and the density values corresponding to each value of the local maxima are obtained. Finally, all local maxima are graded following their density values, and the local maximum values of different density levels can be obtained [32].

## Analysis of the driving forces of industrial evolution

Driving force analysis has long been an important analytical approach. When early geographers studied the driving forces of land use change, they used driving forces to refer to the main factors that lead to changes in land use patterns and divided them into natural physical factors and socio-economic factors [33, 34]. Driving force factors in current studies are often more broadly defined. Natural physical factors mainly include elevation, slope, climate, latitude, and longitude [35–38]; socio-economic factors include population, distance from urban boundaries, distance from roads and railways, road density, and other driving force types [35–42]. Various research methods are used, including geographic detectors and spatial regression models, logic regression models, and global Moran's *I*. The research data are mainly derived from various statistical yearbooks, population and economic census data, and master plans in different periods.

 Scale can have a substantial effect on the results; a scale excessively large reduces the ability to detect important small-scale patterns, whereas a scale too small prevents large-scale patterns from being detected [43, 44]. The landscan population data and GDP grid data are usually on the 1-km scale, and studies conducted on the 1-km grid are also commonly conducted at the provincial scale [45–47]. Thus, the classified distribution of different manufacturing industries was projected on a 1 km×1 km grid in this study, and the number of points of different types of manufacturing was counted in each grid. Higher numbers of points correspond to higher numbers of a certain type of industry somewhere in space. Thus, the evolution process of the industrial pattern is expressed by the difference in the spatial distribution of manufacturing in different periods. A positive difference indicates that a certain manufacturing industry has increased in a certain space, whereas a negative difference indicates the opposite. Finally, an ordinary least squares regression model is constructed according to various natural physical and socio-economic factors that may potentially affect industrial development (Eq 2):

$$\Delta Y = AX_1 + BX_2 + CX_3 + DX_4 \ldots \tag{2}$$

where $\triangle Y$ is the degree of change in the number of industries in each grid, and *X* represents the various factors that may potentially affect the development of manufacturing in each grid.

## Case study

### Study area and data collection

Jiangsu Province is a major manufacturing province that is located in the eastern coastal area of China and has a total area of approximately 107,200 km2. In 2020, Jiangsu's per capita GDP reached USD $20,000, which was second only to municipalities directly under the Central Government of Beijing and Shanghai. It is an important part of the Yangtze River Delta city cluster and shows the highest level of comprehensive development in China [48] (Fig 1). The "Thirteenth Five-Year" Modern Industrial System Development Plan of Jiangsu Province noted that during the "Thirteenth Five-Year Plan (2015–2020)" period, Jiangsu Province's industrialization was in its middle and late stages, and the manufacturing industry has shown clear comparative advantages. In 2015, the province had more than 20 provincial-level advanced manufacturing bases, and the output value of leading industries, such as the electronics, machinery, petrochemical, metallurgy, textile, and light industries, reached a trillion RMB. The output value of the electronics, machinery, and textile industries ranked first in China. Emerging industries such as medicine are developing rapidly, and they have become some of the largest in China.

Amap's POI data in January 2015 and October 2020 (the numbers are 3,276,501 and 6,782,290, respectively) were obtained for Jiangsu Province, and the data were cleaned in SQL (the numbers after cleaning were 605,469 and 695,920, respectively). Afterwards, the domain knowledge base

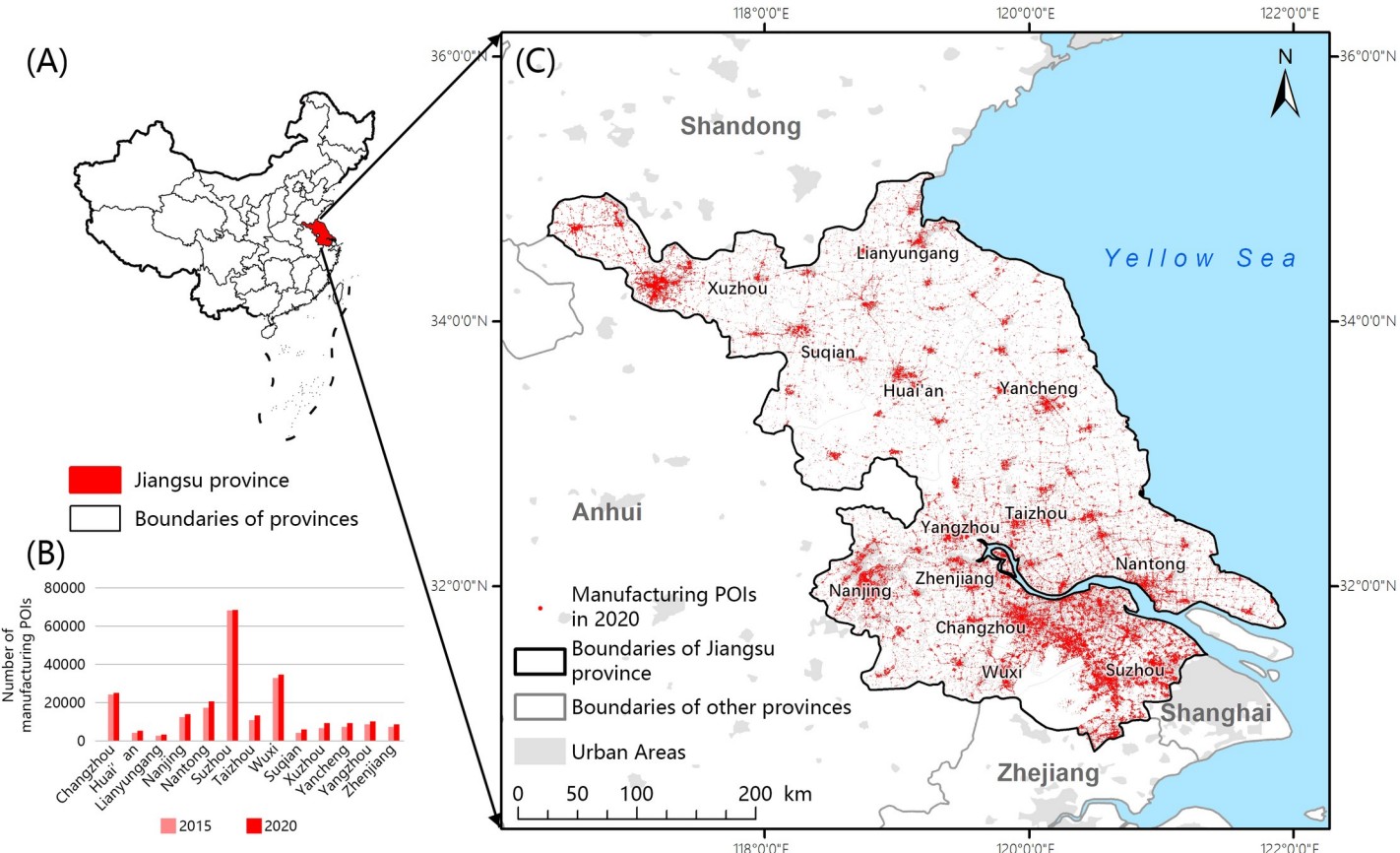

**Fig 1. Distribution of manufacturing POIs in Jiangsu province in 2020.** (A) The position of Jiangsu province in China. (B) Number of manufacturing POIs in cities of Jiangsu province. (C) Distribution of manufacturing POIs in Jiangsu province. The maps were generated by ArcGIS 10.5 and were for illustrative purposes only.

construction and classification methods were used to separate the manufacturing POIs in Jiangsu Province (214,452 and 233,289, respectively, after classification). We then obtained the distribution map of manufacturing POIs in Jiangsu Province in 2015 and 2020. The change in the number of manufacturing POIs in each city in Jiangsu Province in 2020 is shown in Fig 1.

## Distribution pattern of various manufacturing industries in Jiangsu province based on POI natural language processing

According to the text classifying methods, the points in the geographic information database imported into the xy-coordinates of the POIs fall into the 1 km×1 km grid of Jiangsu Province. By deleting POIs with a name similarity greater than 60% in a grid, double counting of the same point in the grid was eliminated, and the seven major types of manufacturing space patterns were shown. A total of 2270 POIs in 230 grids were tested through field investigation. There were 2003 points with a correct rate of 0.88, among which 1831 points were classified correctly. The accuracy was 0.915, and the kappa coefficient was 0.872, which verified the reliability of the classification results. Figs 2 and 3 show the classification results of various manufacturing industries in 2015 and 2020, respectively.

## The evolution of various manufacturing industries during the 13th five-year plan period in Jiangsu province

Based on the distribution maps of the seven types of manufacturing industries, the data in 2015 and 2020 can be used to superimpose the number and space of the industrial distributions of various manufacturing industries. At the same time, the density field hotspots of the seven industries in the two periods of data were created in ArcGIS to quantitatively measure the clustering locations of various manufacturing industries.

## Research on the driving forces of the evolution of different manufacturing patterns in Jiangsu province

According to the analysis of the driving forces, manufacturing base, distance from cities and towns, population, GDP per capita, distance from railway stations, distance from industrial parks, and longitude were the driving factors examined in the analysis of the evolution of the manufacturing industry. Relevant data on driving force analysis factors in Jiangsu Province on a 1-km grid were collected. Table 1 shows various data sources and descriptive statistics. Using Eq (2), the correlations between the degree of change in each manufacturing industry and driving forces were determined (Table 2). All regression models were statistically significant ($p < 0.001$) and had adjusted $R^2$ values ranging from 0.052 to 0.985. Multicollinearity among all regression models was low, and variance inflation factors were less than 1.574 [49–51]. The regression models suggest that various driving forces and related factors have contributed to the development of various manufacturing industries (Table 2).

## Discussion

### Distribution of seven major types of manufacturing industries in Jiangsu province

The natural language processing algorithm proposed in this paper can realize the classification of POI names. By using different classification standards, the companies can be subdivided to varying degrees depending on the goals of the research, which aids research in large-scale economic geography with micro-level data. Using this method, the distribution characteristics of the seven major types of manufacturing in Jiangsu Province can be drawn (Figs 2 and 3).

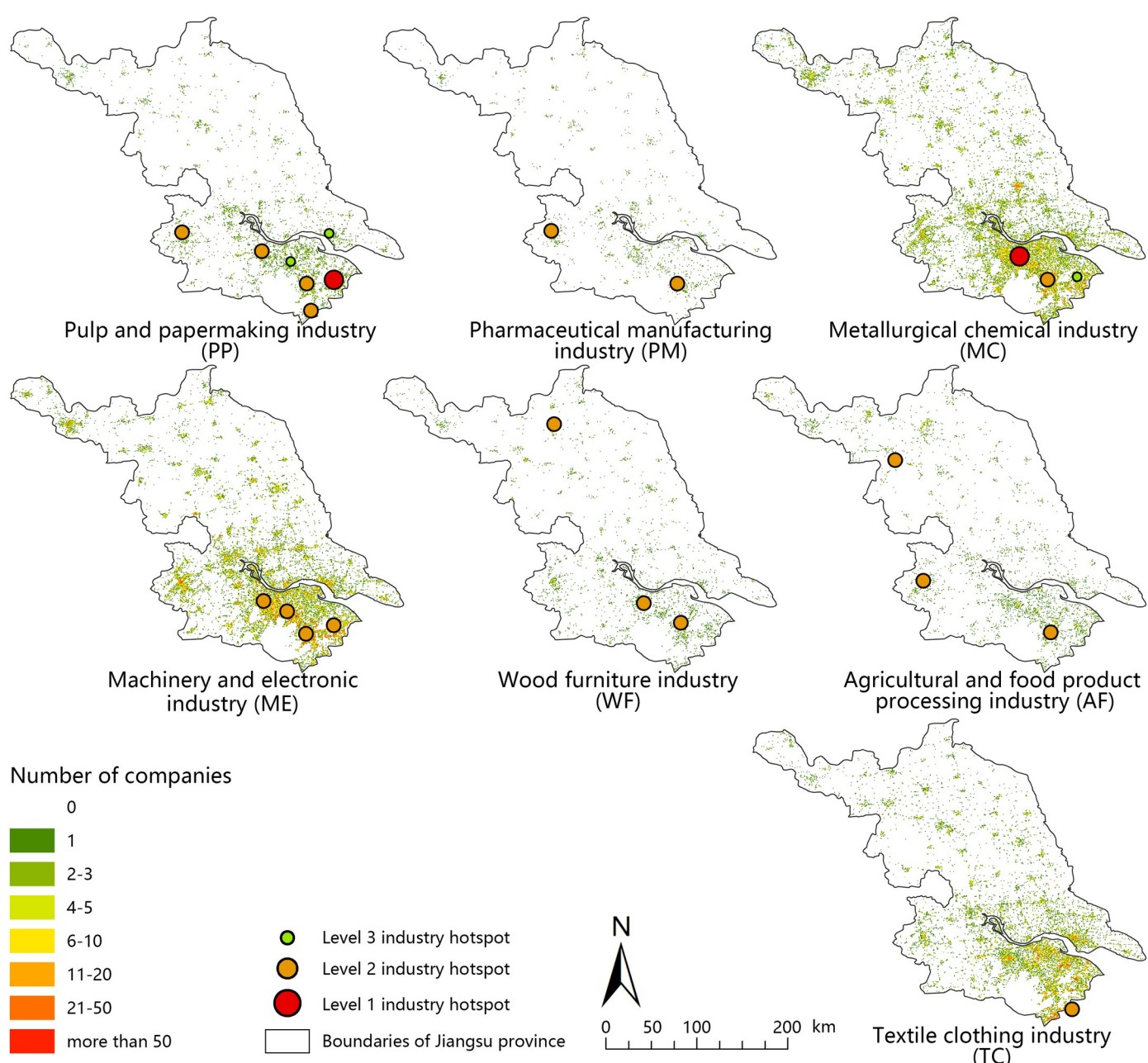

**Fig 2. Distribution pattern of various manufacturing industries in Jiangsu province in 2015.** Number of different industries in a 1 km×1 km projection grid in 2015. Hotspots in 2015 were calculated based on manufacturing POIs distribution. The maps were generated by ArcGIS 10.5 and were for illustrative purposes only.

Among the low-end manufacturing industries, the AF, WF, and PP industries are distributed in both southern Jiangsu and northern Jiangsu, and the TC industry is mainly distributed in southern Jiangsu. Among the traditional manufacturing industries, the MC and ME industries are mainly distributed in central Jiangsu and southern Jiangsu. These two industries are also the largest in Jiangsu Province. The overall number of PM companies is small, and they are distributed in southern Jiangsu and northern Jiangsu, mainly in Nanjing, Suzhou, and Changzhou in southern Jiangsu.

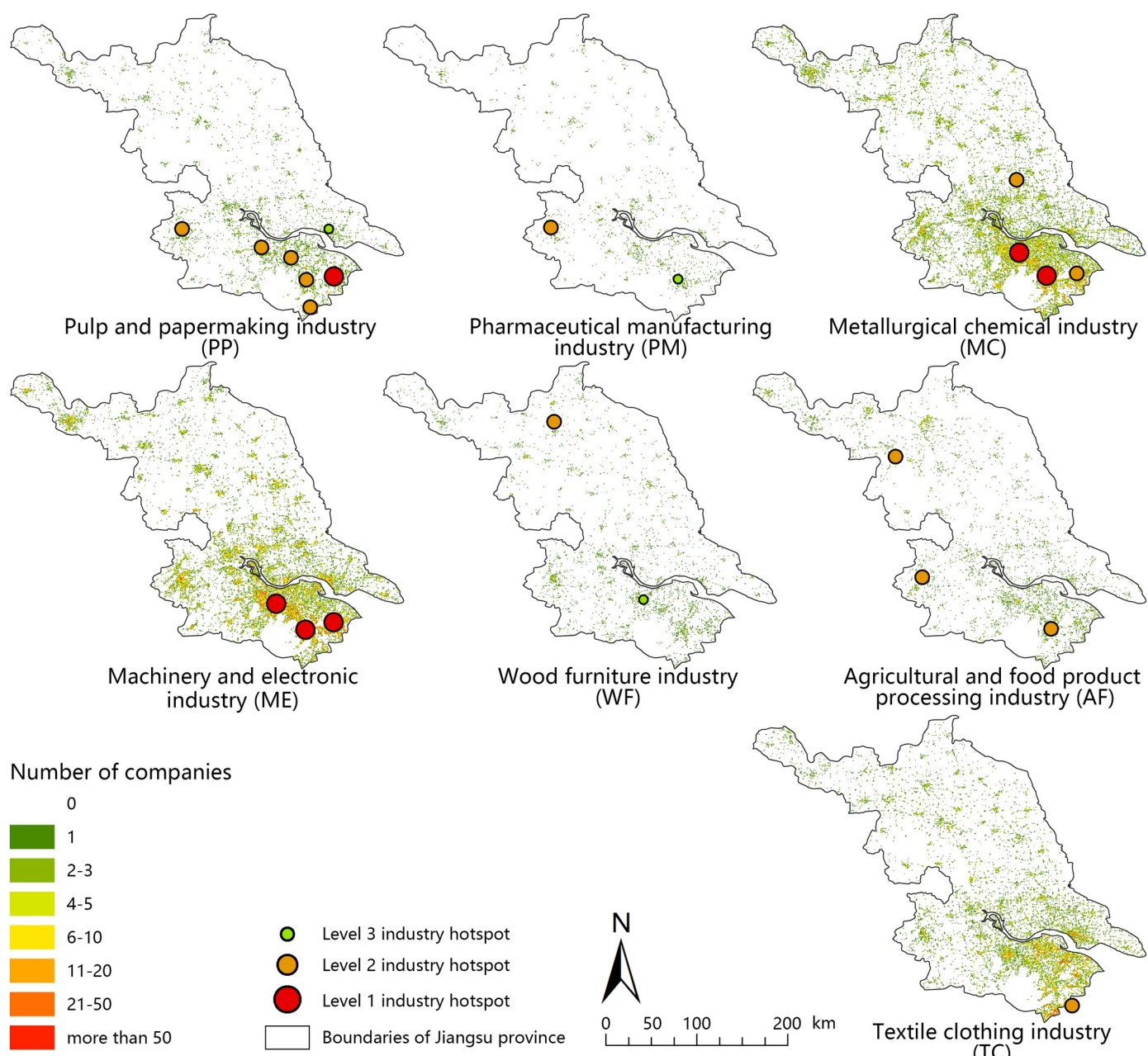

**Fig 3. Distribution pattern of various manufacturing industries in Jiangsu province in 2020.** Number of different industries in a 1 km×1 km projection grid in 2020. Hotspots in 2020 were calculated based on manufacturing POIs distribution. The maps were generated by ArcGIS 10.5 and were for illustrative purposes only.

## Evolution of the seven types of manufacturing industries during the 13th five-year plan

Based on the two-phase manufacturing industry distribution pattern, the most obvious changes were observed for the ME, MC, and AF industries. The changes in PM, TC, and PP industries were not pronounced.

**Table 1. Data sources and descriptive statistics of various industry driving forces.**

| Type | | Code | Variable Name | Calculation Method | Min | Max | Average | Std. |
|---|---|---|---|---|---|---|---|---|
| **Dependent variables** | | $\triangle Y_a$ | $\triangle$TC | Calculation by grid increment | -47 | 124 | 0.03 | 0.847 |
| | | $\triangle Y_b$ | $\triangle$ME | | -727 | 125 | 0.37 | 4.734 |
| | | $\triangle Y_c$ | $\triangle$WF | | -173 | 19 | -0.87 | 4.063 |
| | | $\triangle Y_d$ | $\triangle$AF | | -26 | 27 | -0.04 | 0.547 |
| | | $\triangle Y_e$ | $\triangle$MC | | -38 | 328 | 0.55 | 2.798 |
| | | $\triangle Y_f$ | $\triangle$PM | | -443 | 36 | -0.64 | 3.203 |
| | | $\triangle Y_g$ | $\triangle$PP | | -44 | 30 | 0.07 | 0.577 |
| **Control variables** | **Manufacturing base** | $X_{1a}$ | 2015 TC | The number of industries falling in each grid fishing net in 2015 | 0 | 609 | 0.47 | 3.603 |
| | | $X_{1b}$ | 2015 ME | | 0 | 162 | 0.87 | 4.118 |
| | | $X_{1c}$ | 2015 WF | | 0 | 22 | 0.06 | 0.422 |
| | | $X_{1d}$ | 2015 AF | | 0 | 28 | 0.04 | 0.294 |
| | | $X_{1e}$ | 2015 MC | | 0 | 328 | 0.60 | 2.850 |
| | | $X_{1f}$ | 2015 PM | | 0 | 42 | 0.03 | 0.277 |
| | | $X_{1g}$ | 2015 PP | | 0 | 30 | 0.09 | 0.546 |
| **Explanatory variables** | **Social Interaction** | $X_2$ | Distance from town | After performing Euclidean distance analysis on the town boundary in GIS, a table is used to display the minimum value of the regional statistics, and the logarithm of the minimum distance from the town boundary is taken | 0 | 10.367 | 6.836 | 3.133 |
| | | $X_3$ | Population | Population of Jiangsu province | 0 | 253510 | 834.388 | 2.2 |
| | | $X_4$ | GDP per capita | Divide the total GDP(2015) by the population each grid | 0 | 91719 | 77.774 | 519.101 |
| | **Social Interaction** | $X_5$ | Distance from railway station | After performing Euclidean distance analysis on the station location in GIS, the minimum value of the regional statistics is displayed in a table, and the minimum value of the obtained distance from the high-speed rail station is taken as the logarithm | 0 | 12.036 | 10.466 | 0.791 |
| | | $X_6$ | Distance from industrial park | After performing Euclidean distance analysis on the station location in GIS, the minimum value of the regional statistics is displayed in a table, and the minimum value of the obtained distance from the industrial park is taken as the logarithm | 0 | 12.194 | 10.450 | 0.662 |
| | **Location** | $X_7$ | Longitude gravity | Longitude of site location | 116.371 | 121.901 | 119.440 | 1.026 |

The overall distribution of the WF and AF industries was relatively scattered and slightly concentrated in southern Jiangsu; differences between urban and rural areas were small; and the distribution was contiguous. The number of hotspots in the WF industry decreased during

**Table 2. Regression analysis of industry driving forces.**

| Independent variable | Model of TC Industry | Model of ME Industry | Model of WF Industry | Model of AF Industry | Model of MC Industry | Model of PM Industry | Model of PP Industry |
|---|---|---|---|---|---|---|---|
| $X_1$ | 0.348**** | 0.617**** | -0.161**** | 0.446**** | 0.997**** | -0.025**** | 0.891**** |
| $X_2$ | -0.018**** | 0.033**** | 0.042**** | 0.067**** | 0.004**** | 0.083**** | 0.006**** |
| $X_3$ | -0.053**** | -0.028**** | -0.169**** | -0.038**** | -0.022**** | -0.104**** | -0.07**** |
| $X_4$ | -0.003 | 0.001 | 0.012**** | 0.009** | 0.001 | 0.012**** | 0.002 |
| $X_5$ | 0.031**** | -0.02**** | 0.152**** | 0.036**** | 0.004**** | 0.111**** | 0.028**** |
| $X_6$ | -0.002 | -0.009** | 0.017**** | 0.007* | 0 | -0.014**** | 0.008**** |
| $X_7$ | -0.03**** | -0.054**** | -0.099**** | 0 | 0.001* | -0.077**** | -0.002 |
| **Adjusted $R^2$** | 0.119 | 0.372 | 0.123 | 0.193 | 0.985 | 0.052 | 0.764 |

Independent variables as defined in Table 1. ****, ***, **, and * indicate significance at the 0.0001, 0.001, 0.01 and 0.05 levels, respectively.

the 13th Five-Year Plan period. The scale of Changzhou and Suzhou hotspots was lower at the end of the 13th Five-Year Plan, and the number of industrial hotspots decreased. The number of hotspots in the AF industry remained stable. Among them, the values of Suqian hotspots increased, and the values of Nanjing and Suzhou hotspots declined. The AF industry migrated from southern Jiangsu to northern Jiangsu.

The MC industries are obviously concentrated in southern Jiangsu. The MC industry shifted to northern Jiangsu. New hotspots appeared in Xinghua, Taizhou in northern Jiangsu. The hotspots of Suzhou, Kunshan, and Wuxi in southern Jiangsu are expanding. Hotspots of Wuxi and Changzhou jointly formed a large-scale hotspot across administrative divisions. Hotspots merged in the mechatronics industry. Specifically, the two previously lower-level hotspots in Wuxi and Changzhou merged into higher-level hotspots. The level of Kunshan hotspots declined, and the level of Yangzhou hotspots increased. The overall trend in the MC industries shifted to the cities of Central and Northern Jiangsu. The difference in the central city was negative, and the number of enterprises decreased; the increase was obvious outside of the traditional central city, especially near the boundaries of administrative divisions.

Overall, the pattern of various manufacturing industries in Jiangsu Province exhibited the following characteristics. 1) Industrial hotspots across administrative divisions are becoming increasingly prominent in southern Jiangsu, especially in Changzhou and Wuxi. 2) The spatial concentration of leading manufacturing industries in Jiangsu Province is becoming increasingly pronounced. Among them, this feature is most obvious for the mechanical and MC industries. 3) All types of manufacturing industries in developed regions expand outward from the central city. The mechanical, electronic, and MC industries in the central city of Suzhou are decreasing, and the central city is expanding outward. The policy of Third Industry Priority is an important factor affecting this spatial pattern.

## Driving forces of industrial development of different manufacturing industries in Jiangsu province during the 13th five-year plan

The TC, ME, AF, MC, and PP industries were positively correlated with the number of industries at the beginning of the period (Table 2); that is, greater numbers of original industries resulted in a greater number of new enterprises. The addition of the new WF industry and the new PM industry was not related to the original industries, which means that the overall changes in these industries were not significant. The goodness-of-fit of manufacturing base was the highest (0.997), indicating that manufacturing base is an extremely important driving force.

The ME, WF, AF, MC, PM, and PP industries were positively correlated with the distance from the town. The TC industry showed the opposite pattern. This makes sense given that the development of the TC industry requires a large amount of labor. The distribution of the TC industry around towns aids the acquisition of labor and access to markets.

All the industries were negatively correlated with population size, and most industries were positively correlated with the GDP per capita. Because manufacturing can release pollutants during the production process, manufacturing companies are typically distributed in urban fringe areas where population densities are lower. In addition, according to land rent theory and industrial location theory, lower prices and convenient transportation in urban fringe areas encourage the industrial use of large areas of land [52].

Transportation conditions are one of the important driving forces of manufacturing [53]. Wu Chuangqing [54] found that transportation infrastructure plays a significant role in promoting industrial agglomeration. The patterns of most types of manufacturing in this paper were consistent with this finding: specifically, the number of companies was positively

correlated with proximity to railway stations. This makes sense given that most of these industries depend on transportation; thus, the convenience of transportation directly affects the income of enterprises.

The AF, WF, MC, and PP industries were positively correlated with industrial parks. Industrial parks provide a place for gatherings that promote the optimization and upgrading of the industrial structure [55, 56]. However, the patterns differed among industries. The TC, ME, and PM industries were negatively correlated with industrial parks.

The shift in manufacturing from the coast to the inland is an important driving force in China [57]. In Jiangsu Province, the change in longitude is a manifestation of this driving force. The FC, ME, WF, and PM industries showed significant shifts from the coast to inland, such as Suzhou, Wuxi to Suqian, Xuzhou, and other places in western Jiangsu.

This paper proposes a research framework based on the driving forces of the spatial distribution of multiple industries. Quantitative analysis of the multiple characteristics that characterize the advantages of regional industrial development and the multiple factors that characterize the supporting conditions of regional development was conducted. Based on the analysis of the above driving factors, various industries in Jiangsu Province show obvious migration from southern Jiangsu to northern Jiangsu. This stems from the fact that the Jiangsu government deploys the Party Central Committee and the State Council on high-quality development and coordinated regional development. Consistent with the "Industrial Development and Transfer Guidance Catalog," support for cities in northern Jiangsu and southern Jiangsu involves inter-regional industrial docking and transfer, which facilitates the balanced development of manufacturing in Jiangsu Province. In developed areas, the influence of industrial parks is slightly positive, which promoted the agglomeration of more industries. Therefore, in addition to some policy-oriented factors, the addition of industrial parks in Jiangsu Province also requires consideration of the type of industry, including whether it is an agglomeration industry, which would be more conducive to future development [19].

## Conclusion

Based on Amap's POI data, this paper proposes a method to analyze the evolution of the urban manufacturing industry based on natural language processing technology. Furthermore, the evolution of the spatial pattern of the manufacturing industry in Jiangsu Province during the 13[th] Five-Year Plan is studied. The main conclusions are as follows. 1) Through the acquisition of Amap's POI data and the construction of the domain knowledge base, the industry type classification algorithm based on the natural language processing of POI names is proposed. The empirical test shows that the accuracy reached 95% and the kappa coefficient reaches 0.872. 2) The seven types of manufacturing industries, including the pulp and paper (PP) industry, metallurgical chemical (MC) industry, pharmaceutical manufacturing (PM) industry, machinery and electronics (ME) industry, wood furniture (WF) industry, textile clothing (TC) industry, and agricultural and food product processing (AF) industry, are drawn through a 1 km×1 km projection grid. Based on this, the analysis of the spatial pattern evolution of various manufacturing industries around the 13th Five-Year Plan period was carried out. The PP, PM, WF, and AF industries were distributed in both northern and southern Jiangsu. The MC, ME, and TC industries were mainly distributed in southern Jiangsu. 3) The manufacturing base was the significant driving factor leading to changes in the number of manufacturing companies within the TC, ME, AF, MC, and PP industries. The distance from towns had a positive effect on all industries except the TC industry. Population size had a negative effect on all industries. GDP per capita had a positive effect on the TC and PM industries. The distance from the railway station had a positive effect on all industries except the ME industry. The

distance from industrial parks had a positive effect on the TC and PP industries. The longitude gravity had a negative effect on the TC, ME, WF, and PM industries.

The analytical method of examining manufacturing spatial pattern evolution based on big data and POI name classification proposed in this paper could be used to further subdivide the industrial spatial type and can accurately distinguish changes in micro-spatial patterns of the manufacturing industry, which differs from traditional methods. The industrial spatial evolution analysis method is more reasonable. This study provides new insight that could be used to guide the developed regions of the manufacturing industry, take advantage of regional advantages, and optimize the spatial layout of the manufacturing industry.

## Supporting information

**S1 Table. POI keyword and industry type knowledge base example.**
(DOCX)

**S1 File. POI data used in this study.**
(CSV)

**S2 File. POI data in grids.**
(ZIP)

## Acknowledgments

We would like to thank Amap for the open API access. And we would like to thank Fu Yanjia, Xu ping, Ma xi for helping us to build the industry type knowledge base example.

## Author Contributions

**Conceptualization:** Shiguang Shen, Chenjing Fan, Lin Zhou.

**Data curation:** Chaoyang Zhu, Chengcheng Wu, Xinran Huang.

**Formal analysis:** Chaoyang Zhu, Chengcheng Wu.

**Investigation:** Shiguang Shen, Chaoyang Zhu.

**Methodology:** Shiguang Shen, Chaoyang Zhu.

**Validation:** Chaoyang Zhu, Chengcheng Wu.

**Visualization:** Chaoyang Zhu, Chengcheng Wu.

**Writing – original draft:** Shiguang Shen, Chaoyang Zhu.

**Writing – review & editing:** Shiguang Shen, Chaoyang Zhu.

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
