## [Decision Letter · Decision Letter 0]

26 Apr 2021

PONE-D-21-09214

Research on the Evolution and Driving Forces of the Manufacturing Industry during the “13th Five-Year Plan” Period in Jiangsu Province of China Based on Natural Language Processing

PLOS ONE

Dear Dr. Fan,

Thank you for submitting your manuscript to PLOS ONE. After careful consideration, we feel that it has merit but does not fully meet PLOS ONE’s publication criteria as it currently stands. Therefore, we invite you to submit a revised version of the manuscript that addresses the points raised during the review process.

We look forward to receiving your revised manuscript.

Kind regards,

Jun Yang

Academic Editor

PLOS ONE

Additional Editor Comments:

Reviewer 1

The research topic is interesting, but I think it falls short of publication standards for the following reasons.

1) Insufficient discussion in the introductory section, insufficient grasp of the frontiers of disciplinary development and the latest literature, and lack of citation of key literature. There is a lack of research review of domestic POI data in China.

2) The references are problematic, and it is suggested that the references should be revised according to a certain standard format. For example, the journal of publication in Ref. 22 is incorrectly labelled; it should be the Journal of Economic Geography, not the Journal of Environmental Science.

3) There is a lack of in-depth discussion of the uncertainty of the data and the errors in the results.

4) The paper uses 5km as the grid unit, which seems too large, and it is suggested that the authors further consider whether it could be made smaller.

5) There are many problems with the English language in the whole paper, and professional revision is recommended.

Reviewer 2

This article studies evolution and driving forces of the manufacturing industry based on A map's point of interest during the “13th Five-Year Plan” Period in Jiangsu Province of China. The article has complete content and accurate data, which could give some inspiration to the layout and optimization of Jiangsu’s manufacturing industry in the future. The specific problems are as follows:

(1) The literature reviews part in the introduction is relatively simple. Authors should review the literatures on the evolution and driving force of manufacturing industry and the application of big data interest points in the research of manufacturing industry.

(2)The selection of driving force index in this article is too simple. In addition to the factors such as the latitude and longitude, the degree of transportation convenience, and the distance from the city boundary, authors need to consider whether there are other important factors that affect the spatial distribution and evolution of manufacturing industry in Jiangsu Province

(3) The deep reasons for the spatial distribution and evolution of the major manufacturing industries in Jiangsu province are needed to be analyzed.

(4) The conclusion is too simple and not comprehensive. In addition, this article needs to summarize the conclusion about the spatial evolution of manufacturing industry.

Journal Requirements:

2. We note that Figures 1-3 in your submission contain map images which may be copyrighted. All PLOS content is published under the Creative Commons Attribution License (CC BY 4.0), which means that the manuscript, images, and Supporting Information files will be freely available online, and any third party is permitted to access, download, copy, distribute, and use these materials in any way, even commercially, with proper attribution. For these reasons, we cannot publish previously copyrighted maps or satellite images created using proprietary data, such as Google software (Google Maps, Street View, and Earth). For more information, see our copyright guidelines: http://journals.plos.org/plosone/s/licenses-and-copyright.

2.1.    You may seek permission from the original copyright holder of Figures 1-3 to publish the content specifically under the CC BY 4.0 license. 

2.2.    If you are unable to obtain permission from the original copyright holder to publish these figures under the CC BY 4.0 license or if the copyright holder’s requirements are incompatible with the CC BY 4.0 license, please either i) remove the figure or ii) supply a replacement figure that complies with the CC BY 4.0 license. Please check copyright information on all replacement figures and update the figure caption with source information. If applicable, please specify in the figure caption text when a figure is similar but not identical to the original image and is therefore for illustrative purposes only.

Reviewers' comments:

Reviewer's Responses to Questions

**Comments to the Author**

1. Does the manuscript provide a valid rationale for the proposed study, with clearly identified and justified research questions?

Reviewer #1: Partly

Reviewer #2: Yes

2. Is the protocol technically sound and planned in a manner that will lead to a meaningful outcome and allow testing the stated hypotheses?

Reviewer #1: Partly

Reviewer #2: Yes

3. Is the methodology feasible and described in sufficient detail to allow the work to be replicable?

Reviewer #1: Yes

Reviewer #2: Yes

4. Have the authors described where all data underlying the findings will be made available when the study is complete?

Reviewer #1: No

Reviewer #2: Yes

5. Is the manuscript presented in an intelligible fashion and written in standard English?

Reviewer #1: No

Reviewer #2: Yes

6. Review Comments to the Author

You may also provide optional suggestions and comments to authors that they might find helpful in planning their study.

Reviewer #1: The research topic is interesting, but I think it falls short of publication standards for the following reasons.

1) Insufficient discussion in the introductory section, insufficient grasp of the frontiers of disciplinary development and the latest literature, and lack of citation of key literature. There is a lack of research review of domestic POI data in China.

2) The references are problematic, and it is suggested that the references should be revised according to a certain standard format. For example, the journal of publication in Ref. 22 is incorrectly labelled; it should be the Journal of Economic Geography, not the Journal of Environmental Science.

3) There is a lack of in-depth discussion of the uncertainty of the data and the errors in the results.

4) The paper uses 5km as the grid unit, which seems too large, and it is suggested that the authors further consider whether it could be made smaller.

5) There are many problems with the English language in the whole paper, and professional revision is recommended.

Reviewer #2: This article studies evolution and driving forces of the manufacturing industry based on A map's point of interest during the “13th Five-Year Plan” Period in Jiangsu Province of China. The article has complete content and accurate data, which could give some inspiration to the layout and optimization of Jiangsu’s manufacturing industry in the future. The specific problems are as follows:

(1) The literature reviews part in the introduction is relatively simple. Authors should review the literatures on the evolution and driving force of manufacturing industry and the application of big data interest points in the research of manufacturing industry.

(2)The selection of driving force index in this article is too simple. In addition to the factors such as the latitude and longitude, the degree of transportation convenience, and the distance from the city boundary, authors need to consider whether there are other important factors that affect the spatial distribution and evolution of manufacturing industry in Jiangsu Province

(3) The deep reasons for the spatial distribution and evolution of the major manufacturing industries in Jiangsu province are needed to be analyzed.

(4) The conclusion is too simple and not comprehensive. In addition, this article needs to summarize the conclusion about the spatial evolution of manufacturing industry.

7. PLOS authors have the option to publish the peer review history of their article (what does this mean?). If published, this will include your full peer review and any attached files.

Reviewer #1: **Yes: **Xue Bing

Reviewer #2: No

---

## [Author Response · Author response to Decision Letter 0]

12 Jun 2021

Dear Editor, 

Thank you for your comments and feedback. We have revised the article to ensure that it meets PLOS ONE's style requirements. Figs 1–3 were generated by ArcGIS 10.5 and were for illustrative purposes only. Below, we describe in detail how we have addressed each comment.

Sincerely,

Fan Chenjing 

Associate Professor in Urban Planning, Nanjing Forestry University

=-=-=-=-=-=-=-=-=-=-=-=-=-=-=-=-=-=-=-=--=-=-=-=-=-

Authors’ Responses to Referees’ Comments

(authors’ responses to each comment below noted by red text)

Reviewer #1 

Comment 1: Insufficient discussion in the introductory section, insufficient grasp of the frontiers of disciplinary development and the latest literature, and lack of citation of key literature. There is a lack of research review of domestic POI data in China. 

Response: Thank you for this suggestion. We have reorganized the content of the abstract and added several citations to the first paragraph, especially studies of domestic and international research on POI and research on industry driving forces. We have also added a paragraph that summarizes current research progress.

Comment 2: The references are problematic, and it is suggested that the references should be revised according to a certain standard format. For example, the journal of publication in Ref. 22 is incorrectly labelled; it should be the Journal of Economic Geography, not the Journal of Environmental Science.

Response: Thank you for this comment. We have ensured that the format of the references is in line with the journal guidelines.

There is a lack of in-depth discussion of the uncertainty of the data and the errors in the results.

Comment 3: There is a lack of in-depth discussion of the uncertainty of the data and the errors in the results.

Response: Discussion of POI data uncertainty and error was not included in our second and third paragraphs. Through field investigations and the Baidu map Street view map, 2270 POIs in 230 grids were tested. There were 2003 points with a correct rate of 0.88, among which 1831 points were classified correctly with an accuracy of 0.915, and the kappa coefficient was 0.872, which verified the reliability of the classification results.

Comment 4: The paper uses 5km as the grid unit, which seems too large, and it is suggested that the authors further consider whether it could be made smaller.

Response: Thank you for your suggestion. Using 5km as a grid unit can represent geographic clustering, but it is indeed too large, especially in the context of other driving force data with more fine-scale data. According to the accuracy of the landscan population data, we have re-scaled Jiangsu Province and converted it to a 1km*1km grid for research. There are many documents showing that this scale is suitable for provincial-level research(MA Yuqi, ZHU Xiufang, LIU Xianfeng, LU Nan. A population spatialization method based on DMSP/OLS night-time light data and weighted multi-geographic factors: the example of Liaoning Province, YANG Xuchao, GAO Dawei, DING Mingjun, LIU Linshan. Modeling Population Density Using Multi-sensor Remote Sensing Data and DEM: A Case Study of Zhejiang Province, XIONG Junnan, WEI Fangqiang, SU Pengcheng, JIANG Yuhong. Research on GDP Spatialization Approach of Sichuan Province Supported by Multi-source Data). We found that the adjusted R2 of some industries (e.g., MC industry) was significantly improved.

The revised text relevant to this comment is located in the Case Study and Conclusion sections of the manuscript.

Comment 5:There are many problems with the English language in the whole paper, and professional revision is recommended.

Response: Thank you for this suggestion. The text has been edited by a Native English Language Editor. 

Reviewer #2 

Comment 1: The literature reviews part in the introduction is relatively simple. Authors should review the literatures on the evolution and driving force of manufacturing industry and the application of big data interest points in the research of manufacturing industry.

Response: See the response to Comment 1 of Reviewer #1. 

Comment 2: The selection of driving force index in this article is too simple. In addition to the factors such as the latitude and longitude, the degree of transportation convenience, and the distance from the city boundary, authors need to consider whether there are other important factors that affect the spatial distribution and evolution of manufacturing industry in Jiangsu Province. 

Response: Thank you for this suggestion. We have reorganized the discussion of the factors affecting industry. We have also incorporated industrial parks, nature reserves, and GDP per capita in the regression analysis, and this is now mentioned in the Case Study section. 

Comment 3:The deep reasons for the spatial distribution and evolution of the major manufacturing industries in Jiangsu province are needed to be analyzed.

Comment 4: The conclusion is too simple and not comprehensive. In addition, this article needs to summarize the conclusion about the spatial evolution of manufacturing industry.

Response: In the regression analysis of the driving forces, we added other factors that might be important, such as industrial parks, nature reserves, and water bodies. In the Case Study, we also considered the effect of policy factors in driving the evolution of industry in Jiangsu Province.

---

## [Decision Letter · Decision Letter 1]

2 Aug 2021

Research on the Evolution and Driving Forces of the Manufacturing Industry during the “13th Five-Year Plan” Period in Jiangsu Province of China Based on Natural Language Processing

PONE-D-21-09214R1

Dear Dr. Fan,

We’re pleased to inform you that your manuscript has been judged scientifically suitable for publication and will be formally accepted for publication once it meets all outstanding technical requirements.

Kind regards,

Jun Yang

Academic Editor

PLOS ONE

Additional Editor Comments (optional):

Accept

Reviewers' comments:

Reviewer's Responses to Questions

**Comments to the Author**

1. Does the manuscript provide a valid rationale for the proposed study, with clearly identified and justified research questions?

Reviewer #1: Yes

Reviewer #2: Yes

2. Is the protocol technically sound and planned in a manner that will lead to a meaningful outcome and allow testing the stated hypotheses?

Reviewer #1: Yes

Reviewer #2: Yes

3. Is the methodology feasible and described in sufficient detail to allow the work to be replicable?

Reviewer #1: Yes

Reviewer #2: Yes

4. Have the authors described where all data underlying the findings will be made available when the study is complete?

Reviewer #1: Yes

Reviewer #2: Yes

5. Is the manuscript presented in an intelligible fashion and written in standard English?

Reviewer #1: Yes

Reviewer #2: Yes

6. Review Comments to the Author

You may also provide optional suggestions and comments to authors that they might find helpful in planning their study.

Reviewer #1: the authors have made progressive revisions, there is no further comments from my side, i think it could be accepted after minor revison on grammar mistakes.

Reviewer #2: An interesting and valuable study protocol, I do not have much too many comments about the studies, and the manuscript is suitable for publication.

7. PLOS authors have the option to publish the peer review history of their article (what does this mean?). If published, this will include your full peer review and any attached files.

Reviewer #1: **Yes: **Bing Xue

Reviewer #2: No

---

## [Editor Report · Acceptance letter]

9 Aug 2021

PONE-D-21-09214R1 

Research on the Evolution and Driving Forces of the Manufacturing Industry during the “13th Five-Year Plan” Period in Jiangsu Province of China Based on Natural Language Processing 

Dear Dr. Fan:

I'm pleased to inform you that your manuscript has been deemed suitable for publication in PLOS ONE. Congratulations! Your manuscript is now with our production department. 

Kind regards, 

on behalf of

Dr. Jun Yang 

Academic Editor

PLOS ONE